# Community Stakeholders’ Perspectives on Intimate Partner Violence during Pregnancy—A Qualitative Study from Ethiopia

**DOI:** 10.3390/ijerph16234694

**Published:** 2019-11-25

**Authors:** Bosena Tebeje Gashaw, Jeanette H. Magnus, Berit Scheib, Kari Nyheim Solbraekken

**Affiliations:** 1College of Health Sciences, Jimma University, Jimma 1355, Ethiopia; 2Faculty of Medicine, University of Oslo, 0316 Oslo, Norway; 3Tulane School of Public Health and Tropical Medicine, New Orleans, LA 70112, USA; 4Department of Public Health and Nursing, Faculty of Medicine and Health Sciences, University of Science and Technology, N-7489 Trondheim, Norway; 5Department of Obstetrics and Gynaecology, St. Olav’s Hospital, Trondheim University Hospital, 7030 Trondheim, Norway; 6Institute of Health and Society, University of Oslo, 0318 Oslo, Norway

**Keywords:** IPV in pregnancy, responses, gender and power, capacity building, support and networking, Ethiopia

## Abstract

Intimate partner violence (IPV) in pregnancy adversely affects the health of women and unborn children. To prevent this, the community responses, societal systems, and structures to support victims of IPV in pregnancy are vital. Objectives: to explore community stakeholders’ perspectives related to IPV in pregnancy in Jimma, Ethiopia, and if needed, create the knowledge base for interventions. Methods: using an exploratory design, this qualitative study had a maximum-variation (multiple spectrum sources) sampling strategy with 16 semi-structured interviews of purposively selected key informants representing different community institutions. Guided by Connell’s theory of gender and power, a content analysis of the translated interviews was conducted using Atlas.ti 7 software. Results: reconciliation between IPV victims and their abusers was the solution promoted by almost all the respondents. There was limited awareness of the adverse impacts IPV in pregnancy has on the health of the woman and the foetus. Despite regular encounters with victims, there is no organized or structured operational response to support IPV victims between the participating institutions. Conclusion: the potential danger of IPV for the mother or the unborn child was not well understood by the members of the studied Ethiopian community. Neither coordinated efforts to support IPV victims nor links among relevant agencies existed. The study demonstrated the dire need of coordinated practical action, changes in current socio-cultural norms, formal training and capacity building, awareness creation, clear intervention guidelines, and facilitation of support networks among relevant institutions in Ethiopian communities.

## 1. Introduction

Worldwide, almost one third of all women have experienced physical and/or sexual violence by their intimate partner [1]. Ethiopia has even higher intimate partner violence (IPV) prevalence, with nearly three in four women reporting lifetime IPV experiences [2], and a prevalence of 15–36% during pregnancy [3,4,5,6,7]. A growing body of evidence highlights the adverse short- and long-term physical, reproductive, and mental health consequences for women exposed to IPV during pregnancy [8]. Furthermore, IPV during pregnancy increases the risk of intrauterine foetal death, preterm birth, and low birth weight [9].

The public response shapes the social climate, and either perpetuates or discourages IPV occurrence [8]. A lack of targeted societal responses leads to tolerance or even justification of IPV [8,9]. When the socio-cultural norms and structures condone IPV, suffering women are often ignored [10,11]. Help-seeking, disclosure, and subsequent utilization of available resources are often hindered by socio-cultural, economic or institutional factors [12]. Deep-rooted gender inequality, discriminatory social values [13] and nurturing patriarchal social norms fostering gendered behaviour, allow IPV to occur.

Institutions operating in these systems may either hold a patriarchal view, or might have a tendency to consider wife abuse as a personal and familial issue rather than as a social and legal problem [14]. Likewise, religious leaders, as a part of the larger socio-cultural structure, may enforce a culture of patriarchy, albeit making public statements denouncing partner violence, and use religious teachings to justify it [15,16].

The main aim of the current qualitative study is to explore the perspectives on IPV in pregnancy of Ethiopian community stakeholders from different institutions. Their unique responses related to IPV in pregnancy may offer clarification of their views on the subject, promoting a deeper understanding, and creating the potential for interventions [17]. 

### 1.1. Theoretical Framework

#### The Theory of Gender and Power

To better understand the community stakeholders’ perspectives on IPV in pregnancy, Connell’s theory of gender and power guided the current study [18,19]. Gender and power are embedded in the individual, interactional and institutional dimensions of most societies [20,21]. As outlined by Connell, the three constructs of gender and power (the structure of cathexis, the structures of the sexual division of labour, and the gendered division of power) exist at the societal as well as the institutional level [18,22]. The gendered division of power at the institutional level is maintained by the abuse of power, authority, and control [22]. The structure of cathexis is also referred to as the structure of social norms and affective attachment or attitudes [22]. Connell identified this structure to address the affective nature of relationships, a structure that defines the culturally normative roles for men and women, which may weaken women’s role and increase the inequality felt by women in a heterosexual relationship. At the societal level, the structure of cathexis characterizes the sexual attachments defining appropriate female sexual behavior. In the area of public health, women are adversely affected by such structures, fostering supportive attitudes of wife beating, regarding violence as a notion of masculinity, and enforcing strict gender roles in the society. All are linked to increased IPV in pregnancy implicating the adverse effects of such structures [3,23,24]. This model was used as a theoretical framework to explore the perspectives of different institutions towards IPV in pregnancy.

## 2. Materials and Methods

### 2.1. Study Design

An exploratory design and qualitative approach semi-structured interviews (SSIs)—was used to gather different community stakeholders’ perspectives regarding IPV in pregnancy.

### 2.2. Study Setting

The study was conducted from November 2015 to April 30, 2016, in Jimma Town, one of the zones in the Oromiya regional state. Jimma is one of the largest cities in Ethiopia, located 352 km southwest of the capital, Addis Ababa. It is a multi-ethnic town, distinguished by different religions, cultures, and languages, with three districts (Woredas) and 13 sub-districts (kebeles), and a population of 177,900 [Central Statistics Agency (CSA), 2015]. The participants were interviewed at their workplace, and 16 interviews were completed.

#### Participants

A copy of the ethical approval from the institutional review boards (IRB) of Norway as well as from Jimma University accompanied a collaboration letter to the head office of each study institution to obtain permission for data collection. Participants representing a wide range of age, profession, religion, education, and responsibility offered insight and included public women’s affairs employees (whose primary focus is gender equality), religious leaders, police officers, judges, and prosecutors (of the governmental institutions), (Table 1).

### 2.3. Data Collection Tools and Procedures

Scripted interview guides were developed. We aimed to capture the participants’ experience with IPV victims, current routine and institutionalized responses, their views on reporting to police officers, whether they were aware of potential medical and/or physical consequences of IPV in pregnancy and any existing collaborative efforts or networking with relevant stakeholders.

Notes were taken during the face-to-face interviews, as the majorities were not willing to tape the interview. The primary investigator (PI) conducted the sessions, and two MSc midwives were note-takers. The objectives of the study were explained. After discussing each question, a debriefing session followed to ensure congruency between the notes and the interviewees’ expressions. All these steps lasted an hour on average. The interview guides were pretested in institutions at other sites with similar characteristics to ensure cultural suitability and clarity. The interviews were conducted in Amharic. The PI and the assistants translated the notes into English after each session. A different person checked a few English translated notes to ensure the accuracy of the text, and no major inconsistencies were identified. Additionally, we revisited two interviewees (religious leaders) to seek clarification or additional information about the issues raised in earlier interviews. 

### 2.4. Data Processing and Analysis

Atlas.ti 7 software was used for analyses [25]. Based on previous research on IPV and the theoretical framework, we interpreted the content of the textual data through a systematic process of coding and recognizing themes and categories (Table 2). Finally, after classifying all notes/scripts into their respective categories and themes, we started describing and interpreting them (Appendix A). Qualitative content analysis was used to explore the manifest of content (what the text says) and the latent content (the interpreted meaning), [25]. The goal of content analysis was to provide knowledge and understanding of the phenomenon under study and the outcome of the analysis is categories or themes describing the phenomenon [26,27]. 

### 2.5. Ethics and Consent:

Ethical approval to conduct the study was obtained from Jimma University College of Health Sciences, IRB, Ref No: HRPGC/305/2015 and the Norwegian Regional Ethical Committee (REK), Ref No: 2015/623/REK Nord, in accordance with the Helsinki Declaration of 1975, revised in 2008. Informed written and verbal consent from the respondents was obtained before data collection. The respondents were fully anonymised, and assured about the confidentiality of their responses, their voluntary participation, and the right to terminate at any time.

## 3. Results

Four major themes emerged from the interviews. Each theme is presented, accompanied by illustrative quotes.

### 3.1. Reconciliation

All participants regarded IPV as very common. They had experience with battered women, both pregnant and non-pregnant. The respondents considered IPV a private issue, happening at all ages, but more concealed among those of older age, accepted and/or tolerated in the community unless severe. Interference by neighbours was discouraged. Our informants preferred advising the victim to be tolerant, seeking reconciliation with the partner or spouse.
“I have met many victims and the solutions given depend on the case, some reconciliation, some to police and others to courts, but there is no guarantee. One lady was killed after reconciled with her abusing husband”. (Women’s affairs representative)

The women’s affairs representatives voiced meeting a minimum of 15–20 pregnant and non-pregnant IPV victims per month each; some had minor, some major and visible trauma. They stated that, depending on the severity of the trauma, they might refer the victim to police, with the power to request examinations to gather medical evidence. Law-enforcing bodies (judges, prosecutors, and police officers) also expressed similar experiences, addressing an average of 35 cases of family problems per week, of which 60–70% were partner conflict issues seen in court. They also stated that there is no systematic recording of IPV (criminal issues in the legal system) separately, it is rather classified as a husband and wife issue, and clustered in “family issues”. The interviewed police officers also stated meeting many victims of IPV, some with major trauma and some minor, a minimum of 12–16 per month. They also underscored why they prefer reconciliation to advising women about their rights, which may involve divorce as a consequence.
“When we advise women about their rights, we fear that it may increase divorce rate as a consequence; we do not support divorce as it is bringing us so many consequences, including children of the divorced family, become more raped, become street children….etc., and it goes vicious circle”. (Male judge)

Many women turn to religious leaders for guidance or support in this culture. Leaders stated while trying to intercede on her behalf in IPV situations, sometimes they faced the partners’ anger, such as blaming, being yelled at and so forth.
“I meet partners in conflict almost every day, I have a characteristics/gift of repairing the marriage, but sometimes the aftermath may come to me (Priest), especially when it comes to those drunk ones”. (Religious leader)

When emphasizing the need for women and families to be protected by their husbands, religious leaders described that women are exposed to many problems because they are physically weaker; men can go anywhere, but women cannot, as they always face the risk of abduction or rape. Our informants stated that men should protect and lead their families because they are the heads of the household. It was also stated that a good woman is a pride for her family and that she should make her husband happy and produce children. The interviewed religious leaders cited religious accounts prohibiting divorce, and emphasized the essence of forgiveness.
“We strongly condemn divorce; our bible prohibited divorce, it is sinful, it should only be death that separate couples; whenever there is a conflict between couples, they have to excuse each other.… Jesus is the lord of forgiveness, and this apology helps us to preserve family and children, generation and our country”. (Religious leader)

### 3.2. Reluctance Involving Police

When addressing whether to report IPV cases to the police or not, several of the participants did not recommend reporting IPV victims to the police unless the victim had sustained severe trauma. One significant reason for not reporting IPV instances to the police was that the husband could become more aggressive and most likely end the relationship and the woman may not want to jail her partner either, especially if he was the only one generating any income. Participants also expressed that police often send victims back to local mediators (religious leaders or elders) to work on reconciliation. 

Furthermore, the representatives of the courts did not support reporting cases of IPV to police because it often would lead to divorce, especially if the woman was pregnant and/or they had children. The respondents regarded the community repercussions of divorce as manifold; the financial burden imposed on the woman; lack of social respect; and the stigmatization of the child(ren) as ‘bastard’. The respondents stated that including the police could exacerbate the conflict. Finally, they stated that the police required evidence, which may not be always easy for IPV victims to produce. The participants also said that judges often do not support reporting the issue of IPV to the police or court unless it is beyond the scope of local mediators and very severe.

However, in contrast to religious leaders, the female police officers were in favour of reporting any IPV cases, stressing that the current system did not take IPV seriously.
“IPV should be reported to the police, but some male police officers do have a problem to deal with this issue; they may even be violent towards their wives. When their victimized wives come to me, I will be taking their history together with my supervisor because I am afraid of male police abusers, and many police officers do not consider IPV as violence”. (Female police).

Similarly, irrespective of their gender, prosecutors were in favour of reporting IPV cases to the police.
“…it should be reported to police, not to let it happen again and take a lesson for others”. (Male prosecutor)

Participants underlined that women’s economic dependence on their partner acts as an obstacle when reporting cases to the police.
*“…**it is challenging, in fact, police or court can be a better solution because most men are afraid to step in the police office, court and be punished. On one hand, police and court may have worst ending because he (abuser) may take a strong decision to end the marriage, while divorce for a housewife with a lot of children…hmmm is difficult; on the other hand, to local mediators, or to us (Imam) may not last long/unsuccessful most of the time”. (Religious leader)*

### 3.3. Limited Awareness of the Consequences and Adversity of IPV in Pregnancy

Most interviewees acknowledged that IPV, mainly physical violence during pregnancy, might affect the health of the mother and foetus in general terms.
“Partner violence during pregnancy, I think it may impact foetal development, and foetus may not be healthy”. (Female judge)
”During pregnancy, beating may cause serious danger for the foetus. ‘Foetus listens what is happening outside’, meaning if she feels happy, it feels the same; if she is disturbed foetus will also be disturbed”. (Religious leader)

The participants in our study regarded pregnancy as a time of stress on the woman, and IPV on top of it can augment such a strain and vice-versa (i.e., IPV increases the stress on pregnancy and pregnancy intensifies IPV). Particularly women’s affairs representatives emphasized how pregnancy affects a couple’s relationship by taking away the husband’s attention from the woman.
“Pregnancy brings many problems to the mother and foetus. The husband may even have left her and turn to other ladies”. (Women’s affairs representative)
“I met a lady, who was attacked by her husband and sent to her family with two children when she came back after a month, she found another lady in her own house”. (Women’s affair representative)

During the interviews, the participants provided brief but comprehensive information about the adverse impact IPV can have on the health status of the women, the foetus and the pregnancy outcome.
“Violence in pregnancy involves death; it may result in bleeding and abortion”. (Female police)

### 3.4. Lack of Coordinated Responses or Strategic Plan Addressing IPV

No notable variations in perspective were observed between the participants of the different institutions when it came to assisting IPV victims in an acute stage. They declared that no strong coordinated or strategic support or network exists among the relevant stakeholders, except for the underutilized referral link opportunity between the police, the women’s affairs representatives and the court.

Women’s affairs representatives reflected that there was no coordinated support system for IPV victims. Whenever victims came with severe and visible injuries, they could be referred to the police and the court. It was also stated that some law-enforcing authorities in the court seemed unwilling to let women’s affairs representatives support the victims and follow the verdict process. Some reported to have been threatened by the abuser, and ‘being a woman’ made it difficult to refer victims and effectively carry out their responsibility.
“We do not have a network, even no security even for us, some men used to come to our office and threaten to kill us”. (Women’s affairs representative)
“Because we don’t have strong relationships with women’s affairs office, depending on the case, but mostly we reconcile many victims with their abusive partner, some will be referred to local mediation”. (Female judge)

A male judge elaborated on how the absence of resources and poor linkages complicated the management of victims.
“I meet many victims, but I met one on my way home; she was term pregnant, evicted from her house, I took her to the local court and link her to women’s affairs, and they let the community contribute some money for her survival as the last resort.”

Female police officers also expressed that there is no strong link within or outside their organizations. Because they are women, they do not even feel safe while trying to help victims. There are no shelters for survivors for a temporarily stay.
“There was this female officer who quarrels with her husband because she took home a woman victim of partner violence who had nowhere to go”. (Male police officer)

The leaders in our study shared many stories of victims turning to and seeking their service. Partnerships with other organizations could potentially be helpful in equipping the various community stakeholders, including religious leaders to link their responses with other relevant agencies. However, religious leaders in our study voiced that they do not refer victims to other support sources. Rather, they strongly recommended couples, to forgive each other and reconcile using religiously informed advice and quoting related verses.
“Jesus is the Lord of forgiveness and forgiveness is infinite”. (Religious leader)

## 4. Discussion

The prevailing male-dominated and religiously anchored traditional values in Ethiopia underscore the family as a private institution. This perspective is reinforced by the societal structures, and gender roles are pervasive in societal and institutional agencies. IPV is widely seen as a private issue in Ethiopia, accepted and tolerated by the society [28,29]. This study is the first comprehensive qualitative study undertaken in Ethiopia addressing community stakeholders’ perspectives on IPV in pregnancy.

### 4.1. Reconciliation as the Main Response

One important finding in our study was that IPV is considered normal and a private family issue. Reconciliation is encouraged at the cost of the woman, her health, integrity, and humanity, encouraging women to tolerate and accept their situation. This is a socially desirable solution in situations of family conflict, as divorce is considered religiously prohibited for women. IPV is highly prevalent in this population [3] and based on the interview data, it is common to encounter or observe victims of IPV, including victims beaten to death. 

Consistent with the Connell’s theory of gender and power, social constructs, including the sexual division of labour, power and social norms, conflict/violence in marriage is taken for granted, and women are expected to be tolerant and sacrificial [30,31], enforcing the tendency to preserve women’s subordination. Other studies also confirm the gender inequitable norms and practices linked to IPV [32,33,34]. Men’s cultural and historical authority further strengthens their general sense of power over their partners, and if this power is threatened, permission to resort to violence [35]. Our study participants affirm traditional gender norms, power imbalance as inherent, and this is enforced by structures embedded in the Ethiopian society across formal and informal institutions.

Another challenge is when IPV victims have children. Having children has been identified as both a potential protective and as a risk factor for IPV. It is also challenging because most research studies indicate that women are expected to stay with their violent partners for the sake of their children [36,37] on the one hand, or they have to leave their violent partners to protect their children [38] on the other hand. However, in our study, having children and/or being pregnant were by our informants considered as the main reason for recommending reconciliation as a remedial action. There is a common proverb said by many Ethiopian communities that ‘a Buffalo is stabbed for the sake of her child’, implying that a woman having children should stay in the relationship at any cost. Such a mindset is imposed on women for many reasons, including expecting women to be a sacrifice as the society is discouraging fatherless (divorced) children. This is underpinning the societal norms and attitudes influencing the responses to IPV in pregnancy. It is also worth noting that for women to stand up for their rights, they need economic power [39], yet most women in Ethiopia are economically dependent on their partners, making it difficult for them to be self-sufficient and have the choice to liberate themselves from their abuser.

The legislation in Ethiopia towards IPV, the 2004 criminal code, criminalizes most forms of violence against women and girls including rape, abduction, female genital mutilation, and early marriage [40,41]. However, the legislative framework in not strong enough to protect survivors from domestic violence. There has so far been no separate domestic violence act or law providing specific civil remedies for survivors, such as the right to obtain protection order, compensation relief, residence order, shelter, or medical benefits. There is an absence of sufficient criminal liabilities for perpetrators [40,42]. This may again enforce survivors to stick with an abusive partner.

### 4.2. Reluctance to Initiate OfficersAction

Our study has also identified that there are few formal and informal resources supporting women experiencing IPV in this Ethiopian community. Other studies are in line with our findings [12,43,44]. However, contrary to the responsibilities of the police, most of the participants in our study do not recommend reporting the IPV incident to them, partly because it does not guarantee the victims’ security. In concert with other studies, our study highlights that police officers may even side with the aggressor or hold a patriarchal view, considering abuse as a personal and family issue rather than a societal and legal problem [14]. This may enhance the gender inequitable norms and encourage such practices in the society [32]. This also illustrates that the responses of IPV in pregnancy by agencies are shaped by existing structural and socio-cultural norms [23].

### 4.3. Limited Awareness of the Consequences of IPV during Pregnancy 

Participants regarded IPV in pregnancy as affecting both the mother and foetus in general terms, stating that mainly physical violence had adverse outcomes on pregnancy. Many studies have documented the effects of IPV on maternal health service utilization [24,45,46], and its multifaceted effect on maternal and neonatal outcomes [47,48]. It is also worth noting that not only physical but also other types of IPV in pregnancy can adversely affect maternal and foetal wellbeing. For instance, women in psychological abuse can be at a higher risk of postnatal depression and increased risk of thinking about harming themselves or their infant [49,50]. Furthermore, IPV has been linked with preeclampsia [51], increased risk of third trimester bleeding [47], homicide, and depression [47,52].

The participants in our study stated that pregnancy might increase the spouse’s extra-marital activity placing women at increased risk for IPV. This is in concert with earlier studies reporting that pregnancy increases a woman’s vulnerability to violence by the partner’s reduced commitment to the relationship. He might be regarding the pregnancy as a limitation of free access to his woman’s body, and regarding it as interfering with her ability to perform her traditional role as a homemaker/caretaker [53,54]. This is in line with the Connell’s sexual division of power and labour. Another study state that when pregnancy is added into an already volatile relationship, the violence can become worse [55].

### 4.4. Lack of Coordinated Responses or Strategic Plan

Effective prevention of IPV requires integration and networking among relevant stakeholders [56]. Effective collaboration and networking can provide a spectrum of services to survivors, including: crisis intervention; safety planning; shelter and transitional housing; a supportive health system including specialized counselling; medical examinations; collection of forensic evidences and referral assistance to survivors; employment; and legal advocacy [57,58]. Remarkably, the participants in our study highlighted a lack of networking, except for the rare/weak referral links among the police, women’s affairs, and the court. The absence of strong links among relevant institutions, coupled with complex socio-cultural sanctions against reporting IPV cases to agenesis, women’s economic dependency, and the absence of system responses would partly explain reconciliation being the most common remedial action enforced by our study participants. This highlights the need for training, capacity building, and clear guidelines to effectively prevent IPV and/or assist pregnant victims.

The presence of informal social networks, where religious leaders are part of the collaborative community responses, have been shown effective to improve access, facilitate the utilization of existing IPV services, and changing norms related to it [58,59]. They may also be crucial in responding to the social, emotional, and spiritual needs of IPV victims. Nevertheless, they reinforced patriarchal ideology and advised victims to tolerate and reconcile with their partners [16]. In fact, the religious leaders in our study expressed that divorce is sinful; indeed, most communities in Ethiopia condemn divorce [29], especially during pregnancy it is regarded as highly unacceptable. Similarly, other stakeholders, including the police and members of the court, promoted mediation with the intention of family peace, family stability, economic security, and religious reasons.

Additionally, female police officers and women’s affairs office representatives felt insecure and unsafe while trying to work with IPV victims because perpetrators had threatened them. Participants in our study illustrated the gender inequitable norms prevailing in the society are shared by male officers; consequently, female agents refrained from action for fear of being attacked by the male perpetrator or colleagues. Here, the gender power is operating at different levels influencing the responses to IPV. The differential gender power men and women have, undermines their professional authority to shift social norms, practices, and beliefs related to IPV. Although women who experience IPV require safety, social support, economic security, housing, and legal protection [60], the lack of gender equity makes obvious the need for a gender-sensitive social structure and a multisectoral response that is integrated n order to prevent the adverse personal and societal impact of IPV in pregnancy.

### 4.5. Study Limitations and Strengths

We secured high-quality data by requesting verifications and/or corrections to the notes at the end of each session, and additional interviews to seek clarification or additional information. We accounted for the researchers’ biases by having two note-takers in addition to the interviewer in the room [61]. The complexity of extracting information on the stakeholders’ perspectives towards IPV during pregnancy across a range of socio-demographic populations; from various sources and/or institutions, was mitigated by using a theoretical framework. The dependability of the data was also maintained by training the research assistants. However, the current study has its limitations. The subject studied is very sensitive, and shaped by the social norms and values, so the participants may not have disclosed the real scenario about the topic, or may have provided biased information. As in most qualitative studies, our findings cannot be generalized to a larger population; however, in accordance with the quality criteria of transferability, we argue that the knowledge obtained could be of value for similar groups and/or contexts.

## 5. Conclusions

IPV in pregnancy was commonly encountered by the Ethiopian community stakeholders in this study. However, the global adverse impact of IPV on pregnancy outcomes was not well understood. The perspectives on IPV in pregnancy seem embedded in many socio-cultural, structural, and economic aspects that erected barriers and prevent pregnant IPV victims from obtaining help. The complexity needs to be considered when designing interventions to address IPV in pregnancy in Ethiopia. Formal and informal leaders often regard IPV as a private issue and recommend reconciliation of IPV victims with their abusers. There is no organized or structured operational response to support IPV victims among relevant institutions in Ethiopia. There is a need for formal training, capacity building, clear protocol or intervention guidelines, increased community awareness about the consequences of IPV in pregnancy, and a coordinated and well-structured operational response to support pregnant IPV victims. 

## Figures and Tables

**Table 1 ijerph-16-04694-t001:** Characteristics of the study participants (*N* = 16).

Participant Characteristics		*n*
Age	18–35	5
	36–50	8
	51 & above	3
Gender	Male	10
	Female	6
Marital status	Married	15
	Single	1
Religion	Orthodox	5
	Muslim	6
	Protestant	5
Level of education	Grades 6–12	1
	Above 12th grade	15
Occupation	Law enforcing bodies (police officers, judges, and prosecutors)	7
	Religious leaders	7
	Women’s affairs	2

**Table 2 ijerph-16-04694-t002:** Sample content analysis of the community stakeholders’ responses (theme 1) of different institutions to intimate partner violence (IPV) in pregnancy based on Connell’s theory of gender and power.

Meaning Units	Condensed Meaning Units	Codes	Sub-Themes
Most prefer to reconcile victims with their attacker, advice to calm down and tolerate or not to divorce Neighbours should not interfere Repairing marriage, make a prayer... (religious leaders) No need to report to police	Reconciliation, tolerance, prayer repairing marriage as a solution A tendency not to report IPV cases to police	Reconciliation is commonly preferred Keeping with the marriage	Reconciliation was taken as the best solution to attenuate IPV (social norms, values and attitude)
Women have nowhere to go Most women are economically dependent on their partner	Most women are economically dependent	Economic dependency	Women’s economic dependency as a barrier not to leave an abusive partner (the structure of the sexual division of labour and power)
Women having many children should not or do not want to leave their children behind We fear the consequences of divorce	Women should stay in abusive relationships for the sake of their children fear of the consequences of divorce	Having children Fear of divorce	Having children and fear of divorce as a barrier not to leave abusive partner, (sexual division of labour, social norms, values and attitude)
Police officers do not take IPV seriously Society accepts partner violence as normal IPV is considered as a family and private issue, while women accept IPV and even defend their abuser Married women are more respected No separate record for IPV in the court office Divorce is sinful Women are expected not to leave their children Children of a divorced family are disrespected by society Women are powerless The patriarchal system favours men Marriage is highly valued Women need protection	Considering IPV as normal and/expected Women are expected to sacrifice valuing marriage Perceiving women as powerless A child of a divorced family is disrespected	Trivializing and/or acceptance of IPV Expecting women to sacrifice Social stigma	The structure of cathexis (Patriarchal views, social norms, values, and attitude)

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
