# Peer review of "Community Stakeholders’ Perspectives on Intimate Partner Violence during Pregnancy—A Qualitative Study from Ethiopia"

_ijerph, 2019, doi:10.3390/ijerph16234694_

Round 1

Reviewer 1 Report

This is such an important study and draws out some of the key institutional barriers and causes of violence against women and their unborn child.

I feel the paper has some areas that need improving, but these are technical issues rather than substance and content.

Formatting issues of the tables are a problem. For example, spaces needed between end of paragraph and start of table, plus central alignment of table content, makes it difficult to read the table content – left justification of main content is needed, headings can be centrally formatted (however, this may not be in line with the journals requirements). I would also suggest the sample size and breakdown of the ‘n’ is also not clearly presented, perhaps merging contents in table 1 and 2 might help and have a column dedicated to ‘n’. Table 3, I feel should be in the results section not in methods, as the content relates to the uncovered themes from the analysis. In saying this, I am confused by the presentation of this data as the authors relate to Table S1, but this is not included for me to review in the paper. The results discuss 4 key themes, but these are not tabled in Table 3, I am confused as to why this is the case. In terms of Table 3, I feel the data needs some re-organisation and re-labelling. If I have understood the data correctly, (without S1, I am not sure I have) I would also advise changing the titles of the headings, “meaning units” I feel should be “themes” or “superordinate themes” and “condensed meaning units” are “subordinate themes” – I would argue that you do not need to include a column of “codes” and I am not too sure what the column “subcategories” relates to. Some minor grammatical/punctuation errors throughout e.g. page 4 no period at end of sentence; or in results section a colon should be used ahead of the quote and at end of sentence; and, quotations from participants are right to be in quotation marks but I don’t think the title of the participant should be – that should be outside of the quote.

Author Response

1.This is such an important study and draws out some of the key institutional barriers and causes of violence against women and their unborn child.

Response: Thank you for your appreciation!

2. Formatting issues of the tables are a problem. For example, spaces needed between end of paragraph and start of table, plus central alignment of table content, makes it difficult to read the table content – left justification of main content is needed, headings can be centrally formatted (however, this may not be in line with the journals requirements).

Response: we appreciate your valuable points and have addressed your concerns to the best of our ability.

3. I would also suggest the sample size and breakdown of the „n‟ is also not clearly presented, perhaps merging contents in table 1 and 2 might help and have a column dedicated to „n‟.

Response: this is an interesting point, and we have addressed your comments. We have one column dedicated to‟ n‟, we deleted table 1, and retain table 2 (now labeled as Table 1) which presents the Socio-demographic characteristics of the study participants.

4. Table 3, I feel should be in the results section not in methods, as the content relates to the uncovered themes from the analysis. In saying this, I am confused by the presentation of this data as the authors relate to Table S1, but this is not included for me to review in the paper.

Response: thank you, based on your suggestion table 3 (now labeled as Table 2) is moved under the result section. Please see Table S1, attached as a supplementary material, which shows the steps we went through in the analysis.

5. The results discuss 4 key themes, but these are not tabled in Table 3, I am confused as to why this is the case. In terms of Table 3, I feel the data needs some re-organization and relabeling. If I have understood the data correctly, (without S1, I am not sure I have) I would also advise changing the titles of the headings, “meaning units” I feel should be “themes” or “superordinate themes” and “condensed meaning units” are “subordinate themes” – I would argue that you do not need to include a column of “codes” and I am not too sure what the
column “subcategories” relates to.

Response: thank you for your suggestion, and reflections on the themes. The results are summarized under four themes. We show sample content analysis of theme 1, i.e, the responses of different institutions to IPV in pregnancy based on Connell‟s 3 theory of gender and power. With regards to your comment that the data needs some reorganization and re-labeling, and having codes column, (on Table 2), because it is how it is done in the Atlas Ti software, while the heading “subcategories” is changed to sub-themes. Please also see Table S1 which was attached as a supplementary material which shows the
procedures/steps of data analysis in Atlas Ti software.

6. Some minor grammatical/punctuation errors throughout e.g. page 4 no period at end of sentence; or in results section a colon should be used ahead of the quote and at end of sentence; and, quotations from participants are right to be in quotation marks but I don‟t think the title of the participant should be – that should be outside of the quote.

Response: thank you. Based on your input, the quotes from participants indicated with quotation marks.

Reviewer 2 Report

This is a well written piece of work.  The main topic is of interest mainly because it presents data from a non developed country, which is not well known to the european  societies. The research on IPV during pregnancy is a common place in many countries and many similarities can be found in the content of the research and results with many other research conducted in other patriarchally structured societies of the developing even the developed ones.

Still, the following comments need to be made in order to improve the content of this work.

Most of the participants (14) come from the patriarchally structured institutions and as such the results can be up to a point predictable. How the authors deal with that situation? It is not clear what Woman affairs is. E.g. is it a govermental institution,  a non governmental one, an NGO.. ? Needs to be clarified. How about institutions come from Educational settings, or Health sector which are also vital for dealing with IPV during pregnancy issues? Why these are not considered as informants to the study? What is the legislation on IPV in the country? There is no relevant reference at all. Connell's theory on gender and power is presented very briefly and vaguely in the text. E.g "the structure  of cathexis" is not clearly explained. In general, it needs to be more analytically mentioned as it is the authors main theoretical approach.   Connell's theory needs also to be more well  connected to the whole of the discussion parts. It is only mentioned in the section of "reconciliation ".  Respondents's different religious dogmas are not clearly explained and a connection with how they treat women accordingly is not stated. This could possibly add some information to the researchers for their discussion parts. Content analysis is explained satisfactorily and literature on that  needs to be added.  It is not clear why ethical approval from Norway is needed.  Respondents educational status grades 6-12 , need to be explained what they mean. Data triangulation needs to be further explained and justified for the purposes of the research. 

Author Response

1.This is a well written piece of work. The main topic is of interest mainly because it presents data from a non developed country, which is not well known to the European societies. The research on IPV during pregnancy is a common place in many countries and many similarities can be found in the content of the research and results with many other research conducted in other patriarchally structured societies of the developing even the developed ones.

Response: Thank you for your appreciation, encouraging further work on the issue.

2.Most of the participants (14) come from the patriarchally structured institutions and as such the results can be up to a point predictable. How the authors deal with that situation?

Response: thank you for your meticulous observation. Yes, reconciliation between IPV victims and their abusers was the solution encouraged by almost all the respondents (i.e, may be predictable from the patriarchally structured institutions). However, our study also identified the responses and context why the different respondents prefer reconciliation differ which is very essential for any intervention.

3. It is not clear what Woman affairs are. E.g. is it a govermental institution, a non governmental one, an NGO.. ? Needs to be clarified.

Response: Thank you for raising this important issue. Women’s affairs are in our context the office under Ethiopian governmental structures/institution working in Jimma town (study site).

4. How about institutions come from Educational settings, or Health sector which are also vital for dealing with IPV during pregnancy issues? Why these are not considered as informants to the study?

Response: we really appreciate your concern. Yes, the Health sector plays a vital role on IPV during pregnancy, but we prefer/plan to study it separately.

5. What is the legislation on IPV in the country? There is no relevant reference at all.

Response: thank you for your observation. We include the legislation on IPV in the Ethiopia with the following references:“The legislation in Ethiopia towards IPV, the 2004 criminal code, criminalizes most forms of violence against women and girls including rape, abduction, female genital mutilation, and early marriage. However, the legislative framework in not strong enough to protect survivors from domestic violence. There has so far been no separate domestic violence act or law providing specific civil remedies for survivors; such as the right to obtain protection order,3compensation relief, residence order, shelter, or medical benefits. There is an absence of sufficient criminal liabilities for perpetrators. This may again enforce survivors to stick with an abusive partner”.

6. Connell's theory on gender and power is presented very briefly and vaguely in the text. E.g "the structure of cathexis" is not clearly explained. In general, it needs to be more analytically mentioned as it is the authors main theoretical approach.

Response: we really appreciate your concern. We have addressed your concern in the text (See on page 1): “Connell identified this structure to address the affective nature of relationships, a structure that defines the culturally normative roles for men and women which may weaken women’s role and increase the inequality felt by women in a heterosexual relationship. At the societal level, the structure of cathexis characterizes the sexual attachments defining appropriate female sexual behavior. In the area of public health, women are adversely affected by such structures, fostering supportive attitudes of wife beating, regarding violence as a notion of masculinity, enforcing strict gender roles in the society. All linked to increased IPV in pregnancy implicating the adverse effect of such structure”.

7. Connell's theory needs also to be more well connected to the whole of the discussion parts. It is only mentioned in the section of "reconciliation ".

Response: thank you. Per your suggestion we have modified and connected Connell’s theory to the whole of the discussion parts (see statements highlighted in red in the discussion section emphasizing the Connell’s theory of gender and power constructs).

8. Respondents’ different religious dogmas are not clearly explained and a connection with how they treat women accordingly is not stated. This could possibly add some information to the researchers for their discussion parts.

Response: thank you. Regarding how religious leaders treat victim women, we have stressed their common advice as tolerating and /or reconciling with the abusers, despite type or severity of the abuse. Divorce is considered sinful and religiously prohibited. This is stated both in the result and discussion section.

9. Content analysis is explained satisfactorily and literature on that needs to be added.

Response: Thank you for your meticulous observation. We have added the requested literature under the analysis section.

10. It is not clear why ethical approval from Norway is needed.

Response: thank you, ethical approval from Norway is needed as the 1st author is Oslo university PHD student and all co-authors and /supervisors are from Norway, Oslo and Trondheim university. This is a collaboration project between Jimma University (in Ethiopia) and Oslo University (Norway).

11. Respondents’ educational status grades 6-12, need to be explained what they mean.

Response: thank you, educational status (grades 6-12), means the participants’ level of education is between 6-12 grades.

12. Data triangulation needs to be further explained and justified for the purposes of the research.

Response: thank you for your concern. We have deleted this term. .
